# The relationship of recreational runners' motivation and resilience levels to the incidence of injury: A mediation model

**Patxi León-Guereño[1], Miguel Angel Tapia-Serrano[2]\*, Pedro Antonio Sánchez-Miguel[2]**

**1** Faculty of Psychology and Education, University of Deusto, Donostia-San Sebastian, Spain, **2** Department of Didactics of Musical, Plastic and Body Expression, Faculty of Teacher Training, University of Extremadura, Cáceres, Spain

\* pesanchezm@unex.es

**Data Availability Statement:** All relevant data are within the paper and its Supporting Information files.

## Abstract

Running participation has increased significantly in the last decade. Despite its association with different health-related aspects, athletes may experience adverse outcomes, including injuries. The aim of this study was twofold: to examine the relationship between runners' resilience levels, motivation and incidence of injury, on the one hand; and to analyse the mediation that intrinsic and extrinsic motivation has on the association between the number of injuries and psychological resilience levels among amateur athletes. The sample consisted of a total of 1725 runners (age: 40.40 ± 9.39 years), 1261 of whom were male (age: 43.16 ± 9.38), and 465 of whom were female (age: 40.34 ± 9.14). Athletes completed the Behavioural Regulation in Exercise Questionnaire (BREQ-3), the Resilience scale (CD-RISC 10), and an Injury retrospective survey. Three mediation models were constructed, and the results showed a significant indirect association of athletes' intrinsic motivation and resilience on the number of injuries ($\beta = 0.022$, CI = 0.007, 0.0) in mediation model 1, whereas extrinsic motivation was found to have no significant association on those variables ($\beta = -0.062$, CI = -0.137, 0.009) in mediation model 2. Model 3 showed significant differences with respect to resilience ($p < 0.05$) and intrinsic motivation ($p < 0.05$). Therefore, the mediation of intrinsic motivation on athletes' resilience levels and incidence of injury was demonstrated, i.e., it was found that intrinsic motivation was associated with a higher incidence of injury, while no such correlation was found for extrinsic motivation. This study shows that the amateur long distance runners with a high level of intrinsic motivation tend to suffer from a greater number of injuries, and at the same time psychological resilience was associated with a lower number of injuries.

## Introduction

Participation in recreational running has increased in recent years in western countries [1,2]. An example of this is participation in the Behobia-San Sebastian half marathon race, where the total number of runners has almost tripled in the past ten years [3], reaching approximately

**Funding:** The author(s) received no specific funding for this work

**Competing interests:** The authors have declared that no competing interests exist

30,000 registrations in 2018. This increase in participation can be explained by the nature of recreational running or jogging, which provides a low-cost option with very few extra expenses. Given the limited free time available to most people in western countries, this sport can be easily adapted to tight schedules and different age ranges [4].

Physical Activity (PA) in general is associated with improvement in physical health [5] and with good mental health [6,7]. Recreational running or jogging, in particular, is also connected to participants' health benefits [4,8–10]. However, overuse injury is more highly associated with running than with any other form of aerobic exercise [11,12] especially among novice runners, a field which has been little researched [1,13]. Working along similar lines, Van Gent *et al.* [14] took into account athletes' experience and used a multifactorial approach to this area. They classified injury risk factors into: personal factors, running/training-related factors and health and lifestyle factors. Adopting a wider approach to injury in high-performance sport, McGregor [15] took into consideration a range of factors such as team environment, facilities, athletes-coach relationship, technical-tactical skills, and different psychological aspects such as confidence, resilience and motivation. Injuries usually cause periods away from the sports practice [16] and beyond physiological and physical attributes associated with athletes' injuries, numerous studies have linked the incidence of sports injuries to some psychological attributes e.g. stress, resilience, etc [16–19].

The psychological aspect that has been analysed most has been the motivational characteristics of athletes, as several studies have been conducted on the basis of Self-Determination Theory (SDT) [20]. In this sense, motivational aspects also help to explain athletes' participation in running practice [21]. Also, motivational aspects have been among the most analysed aspects within athletes' personal factors, and have been associated with multiple variables, including motives for participation [21], resilience [22], and performance in endurance races [23], focusing mostly on intrinsic and extrinsic motivation [19]. The positive impact of motivation on runners' health status has also been shown [21,24], and motivation has been analysed as part of the psychological factors that affect return-to-play decisions made by athletes after injuries [25,26]. However, the relationship of this variable to athletes' physical injuries has not been specifically addressed yet and remains unclear.

Psychological resilience (Resilience is defined as the human ability to adapt in the face of tragedy, trauma, adversity, hardship, and ongoing significant life stressors), or the ability to overcome adversity, has been analysed in a variety of contexts [27]. Due to the importance that it has gained over the past decade, this construct has started to be studied in the sports context too [28–31]. Most definitions of resilience encompass two core concepts: adversity and positive adaptation. Psychological resilience was defined by Fletcher and Sarkar [29] as '*the role of mental processes and behaviour in promoting personal assets and protecting an individual from the potential negative effect of stressors*' (p.16). Specifically referring to the sports context, Secades *et al.* [31] defined resilience as the capacity to face and positively adapt to highly stressful situations, which is influenced by personal and contextual aspects [32,33].

Resilience research in sports settings has focused mainly on competitive athletes of different sports [34] on seeking improved performance by individuals [22,28,30,35–38] and teams [39–41] and on how resilience is connected with contextual factors [42,43]. In recent studies, resilience has been associated with elite athletes' injuries [16,33,44], and even though some research has linked this construct to athletes' competitive level [45], very little has been explored in relation to amateur athletes [43] or connecting this psychological factor with non-professional athletes' injuries.

Current trends agree that resilience is a multifactorial and highly complex concept [32,33], which has been lately associated with athletes' injuries [16]. However, sports people's injuries have not been associated with the internal and external motivations to date. In light of this,

and due to the scarcity of research related to amateur athletes' psychological characteristics and injuries, and taking into account that women's participation is significantly lower at the Behobia-San Sebastian race, and the number of research is lower for women [3] differences between male and female athletes will be considered, and the following objectives were formulated for this study: (1) to examine the relationship between runners' resilience levels, internal and external motivation and incidence of injury; and (2) to evaluate the mediation role of intrinsic and extrinsic motivation on the number of injuries and resilience levels of amateur runners. In line with these objectives, the first hypothesis was that the number of injuries would be negatively associated with the resilience level of athletes. Hence, those runners with greater psychological resilience would be related with a fewer number of injuries. The second hypothesis was that intrinsic motivation would have a greater incidence on athletes' resilience than extrinsic motivation, and that it would be positively associated with the number of injuries suffered by athletes. Thus, it was hypothesised that resilience levels could be explained by greater intrinsic motivation, and this relationship would show a higher number of injuries in athletes.

## Material and methods

### Design and participants

This is a descriptive, quantitative, cross-sectional study. The sample was chosen randomly from among the participants of the 54th Behobia- San Sebastián half marathon. The athletes who ultimately did not take part in the race were excluded (Fig 1). The questionnaire was completed by 1725 runners (age: 40.40 ± 9.39 years), 73% of whom ($n$ = 1260) were male (age: 43.16 ± 9.38 years) and 27% of whom ($n$ = 465) were female (age: 40.34 ± 9.14 years), and all of them provided written informed consent to participation in the survey.

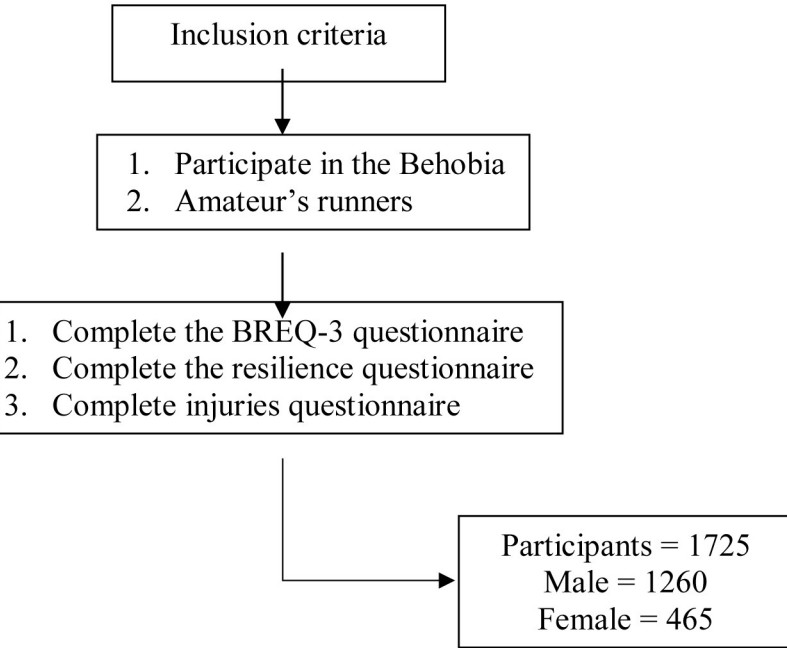

**Fig 1. Flow chart of participants according to inclusion criteria.**

## Instruments

**Regulation of behaviour in physical exercise.**   We used a modified version of the Behavioural Regulation in Exercise Questionnaire (BREQ-3; [46,47]), which includes 6 styles of behaviour regulation: amotivation, external, introjected, identified, integrated, and intrinsic. Each regulation style has 4 items, rated on a 5-point Likert-scale (1 = 'not true for me', 3 = 'sometimes true for me', 5 = 'very true for me'). The BREQ-3 supplies mean scores for each regulation subcategory, where each subcategory reflects the continuum of self-determination (or autonomous behaviour).

**Resilience.**   Resilience was evaluated using the short version of the CD-RISC, in particular, the Spanish adaptation by Notario-Pacheco *et al*. [48] It is made up of 10 items (those numbered 1, 4, 6, 7, 8, 11, 14, 16, 17 and 19) from the original scale developed by Connor and Davidson. [49] Questions are ordered from one to ten (e.g., 'I can deal with any situation') in a five-point Likert scale, where the lowest score indicates 'never' and the highest score means 'almost always'. Consistent with past research [32,50], a sum score was calculated for the analysis. The CD-RISC has been shown to have good reliability (α = .88 and .89), test-retest reliability (.87), and convergent and divergent validity in the development of the scale [49,50]. Cronbach's alphas for the various CD-RISC versions in the current study were .77 for the 5-factor CD-RISC, .89 for the unidimensional CD-RISC, and .87 for the CD-RISC-10.

**Injuries questionnaire.**   To assess the incidence and type of injuries sustained by runners, a retrospective questionnaire [51] was used that was adapted from a previously validated survey [52]. Athletes were labelled as 'injured' or 'non Injured' at the time of the data collection, and a record was made of the number of injuries they had sustained over the preceding twelve months [52]. The tool was changed to make it suitable for this specific context, and included some sociodemographic questions (e.g., place of birth) and questions concerning athletes' physical characteristics (e.g., weight, height) and expected performance, among other aspects. The initial question 'Have you ever suffered from an injury while running training?' was followed by a more detailed battery of questions to define the region and characteristics of the injury.

## Procedure

Approval was obtained from the Ethics Committee of the University of Extremadura, Spain. The study was consistent with the Helsinki declaration of 1975. Participants were treated ethically under the American Psychological Association ethics code [53] regarding consent, anonymity and responses. Participants were contacted by email two weeks before the race, and provided with a detailed introduction to the study and with a link to enable them to access the survey. The questionnaire was created using Google Docs technology [54] and was tested prior to its release. At the end of 2017, Fortuna Sports club and Behobia-SS organisers were contacted and presented with the research project. A formal document specifying the research aims and the different variables and tools to collect the data was also provided to them. Several meetings were held in 2018 to adjust the different measurements of recreational runners' characteristics. Data collection was carried out through Behobia-SS organisers, who emailed a link to the online survey to the 28,737 runners who took part in the race. The confidentiality and anonymity of the data was guaranteed to the participants, who were also informed that the data collection would be used for academic purposes and in order to improve the Behobia-SS event. The participating athletes were provided with an email address that they could contact for any queries they might have while completing the survey.

## Statistical analysis

Firstly, descriptive statistics were shown, including means and standard deviations (SD). The differences between data by sex were calculated using the student's $t$ test and the chi-square test for continuous and nominal variables. The partial correlations controlled for by sex and age were calculated to analyse the relationships between resilience, intrinsic motivation and extrinsic motivation. The confidence interval obtained after the analysis of the covariance of the dependent variables showed $p < 0.20$.

The association of number of injuries with resilience and motivation indicators was tested using a linear regression model. Linear regression analyses were performed and adjusted for covariates by creating 3 models. Model 1 was further adjusted for age, sex and resilience; Model 2 was additionally adjusted for intrinsic motivation, and Model 3 was further adjusted for extrinsic motivation.

The PROCESS SPSS version 3.00 Macro was used for the mediation analysis between number of injuries and resilience. Two different models were established. For mediation model one, extrinsic motivation was specified as a mediating variable; resilience was used as an independent variable and the number of injuries was used as a dependent variable. For mediation model two, the dependent and independent variables were maintained, but the mediating variable was intrinsic motivation. The mediation hypothesis was tested using the bootstrap method, based on 10,000 bias-corrected 95% confidence intervals. The point estimate was considered significant when the confidence interval (CI) did not contain zero. The remaining statistical analyses were performed using SPSS version 3.0 for Windows, and the level of significance was set at $p < 0.05$.

## Results

Table 1 shows the descriptive statistics of the study sample. In general, male participants presented lower levels of intrinsic motivation ($p < 0.01$) and a greater number of injuries than female participants ($p < 0.001$). However, no significant differences were found for age, resilience and intrinsic motivation ($p > 0.05$).

Table 2 shows the associations between number of injuries and resilience and motivational indicators. In model 1, the number of injuries did not show a significant association with resilience levels after controlling for sex and age ($\beta = -0.063$; $p = 0.089$). However, in model 2 the number of injuries was positively associated with intrinsic motivation after controlling for sex, age and resilience ($\beta = 0.123$; $p < 0.05$). On the other hand, there was no significant association between the number of injuries and extrinsic motivation after additional adjustment for intrinsic motivation in model 3 ($\beta = -0.102$; $p = 0.072$).

**Table 1. Descriptive characteristics of the study sample and differences by sex.**

|  | All participants | | Males | | Females | | *P* |
|---|---|---|---|---|---|---|---|
|  | **M** | **SD** | **M** | **SD** | **M** | **SD** | |
| *N* | 1725 | | 1260 | | 465 | | |
| Age | 42.40 | 9.39 | 43.16 | 9.38 | 40.34 | 9.14 | 0.871 |
| Resilience | 4.05 | 0.55 | 4.04 | 0.55 | 4.05 | 0.55 | 0.780 |
| Intrinsic motivation | 4.63 | 0.51 | 4.61 | 0.52 | 4.70 | 0.47 | **0.003** |
| Extrinsic motivation | 3.25 | 0.37 | 3.23 | 0.37 | 3.31 | 0.35 | 0.448 |
| Injuries | 1.21 | 0.86 | 1.26 | 0.86 | 1.06 | 0.84 | **0.000** |

**Table 2. Associations of number of injuries with resilience, intrinsic motivation and extrinsic motivation.**

|  | Number of injuries | | | | | |
|  | Model 1 | | Model 2 | | Model 3 | |
|  | β | p | β | p | β | p |
|---|---|---|---|---|---|---|
| Age | 0.013 | **0.000** | 0.013 | **0.000** | 0.013 | **0.000** |
| Sex | -0.169 | **0.000** | -0.180 | **0.000** | -0.173 | **0.000** |
| Resilience | -0.063 | 0.089 | -0.085 | 0.023 | -0.088 | **0.019** |
| Intrinsic motivation |  |  | 0.123 | **0.003** | 0.140 | **0.001** |
| Extrinsic motivation |  |  |  |  | -0.102 | 0.072 |

p: significance level. β: Values are standardised; Model 1: Age + Sex + Resilience; Model 2: Age + Sex + Resilience + Intrinsic motivation; Model 3: Age + Sex + Resilience + Intrinsic motivation + Extrinsic motivation

The results of the simple mediation model 1 (Fig 2: Model 1) showed a significant indirect effect of resilience levels on the number of injuries (β = -0.062, 95% $SD$ = 0.036, 95% CI [-0.137, 0.009]). There was a negative correlation between the number of injuries and resilience levels (β = -0.062, $t$ = -1.695, $p > 0.05$), and between the number of injuries and extrinsic motivation (β = -0.058, $t$ = -1.056, $p > 0.05$). However, there was a positive correlation between extrinsic motivation and resilience levels (β = 0.002, $t$ = 0.125, $p > 0.05$). The results obtained

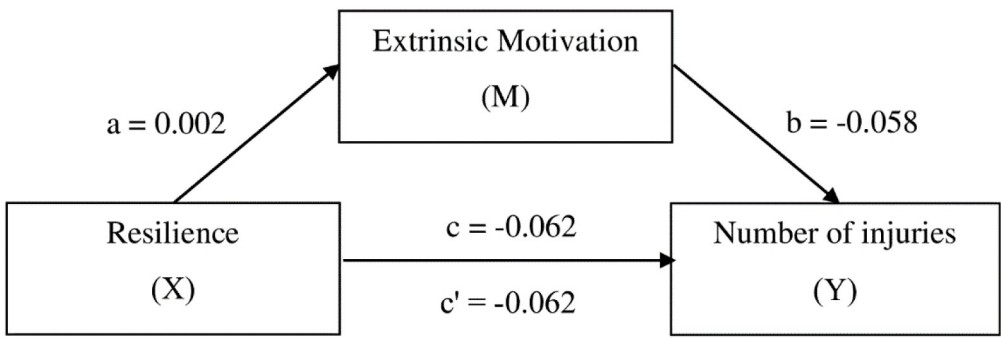

IE: β = -0.062, $SD$ = 0.036, 95% CI [-0.137, 0.009]

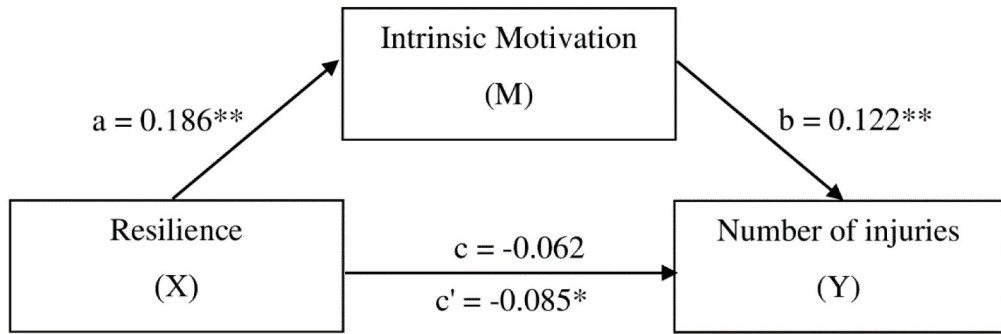

IE: β = 0.022, $SD$ = 0.082, 95% CI [0.007, 0.0]

**Fig 2. Mediation model.**

in the simple mediation Model 2 (Fig 2: Model 2) showed a significant indirect effect of resilience on the number of injuries (β = 0.022, *SD* = 0.082, 95% CI [-0.007, 0.009]). There was a positive correlation between the number of injuries and resilience levels (β = -0.085, *t* = -2.277, *p* > 0.05). There were also positive correlations between the number of injuries and intrinsic motivation (β = 0.122, *t* = 3.014, *p* < 0.001), and between intrinsic motivation and resilience levels (β = 0.186, *t* = 8.566, *p* < 0.001).

## Discussion

The aim of this study was to analyse how psychological resilience and intrinsic and extrinsic motivation are associated with the number of injuries suffered by amateur runners. The mediating effect of intrinsic and extrinsic motivation on the variables indicated above was also analysed. The results showed a positive correlation between athletes' intrinsic motivation and their number of injuries, such that the higher the athletes' intrinsic motivation, the higher their number of injuries. At the same time, athletes' resilience was negatively related to injuries, which resulted in high psychological resilience being significantly associated with fewer injuries suffered by runners. However, the effect of extrinsic motivation did not show a significant relationship to the number on injuries. Therefore, it was concluded that resilience level of amateur runners was associated with suffering fewer injuries, and that athletes who were more intrinsically motivated were more likely to have higher number of injuries.

The results obtained by mediational models 1 and 2 revealed that extrinsic motivation and intrinsic motivation have a positive effect on the relationship between resilience and the number of injuries. In both cases, a negative correlation was shown between athletes' resilience and number of injuries, thus confirming the first hypothesis of this study. In this line, Zurita *et. al* [19] and Abenza *et. al* [17] showed the influence of stressors and resilience on athletes' injuries, and emphasised the importance for athletes to control stress and their relationship with the capacity to overcome these adverse outcomes. According to our findings, the more resilient athletes were, the better they were able to manage stressful situations, which may provide an explanation on why these athletes suffered fewer injuries than athletes with a lower level of psychological resilience. These characteristics have been shown to vary depending on athletes' competitive level [45]. For example, Jaenes et al. [37] and de la Vega, Rivera, and Ruiz [38] described high-performance endurance runners as having a hardy personality, and Sarkar and Fletcher [33] also analysed athletes' resilience in connection with performance. Moreover, athletes' resilience and injuries have been studied in different competitive levels [45] and especially in high-performance athletes [16,44]. However, these psychological characteristics have been examined in relation to overcoming injury processes, i.e. recovery after injury, but not as an attempt to explain the relationship between these psychological variables and the incidence of injury.

The results also showed that the intrinsic motivation variable acts as a mediator of the relationship between athletes' resilience and their number of injuries to a greater extent than extrinsic motivation does, thus confirming the second hypothesis of this study. Model 2 revealed that there were statistically significant associations among the different variables: Intrinsic motivation with resilience levels (β = 0.186, *t* = 8.566, *p* < 0.001) and the number of injuries (β = 0.122, *t* = 3.014, *p* < 0.001). From these results it can be concluded that the more intrinsically motivated athletes are, the greater the number of injuries they may experience. This may be due to the fact that athletes who are more self-determined towards PA are more likely to strive for self-improvement, and at the same time tend to push their boundaries further, which could lead them to have more injuries. Previous research has shown how high-performance endurance athletes present a high level of intrinsic motivation [23], but the higher

the athletes' performance level, the higher their capacity of being resilient and facing stressful situations. High-performance athletes have been found to be more capable to face different setbacks in a more effective way [45].

The study has some limitations. Some variables such as the training load and training programmes followed by athletes that were not taken into account as potential elements influencing motivation and incidence of injury. Likewise, another limitation was that the information on injuries was self-reported. The cross-sectional study design was another limitation, as it does not allow establishing causal relationships between the study variables. Despite these limitations, very few studies have analysed the association between psychological variables and the incidence of injury to date, and as far as we are aware, no research has focused so far on analysing motivational and resilience variables in connection with the number of injuries suffered by athletes. In line with Sánchez *et al.* [45], it would be useful for future research to analyse motivational aspects in different performance levels and in different sports, with regard to the incidence of injury and the psychological resilience of sportspeople when overcoming adverse situations. This would show to what extent athletes' number of injuries relates to their resilience levels. In addition, controlling for training frequency and duration, as well as nutrition, hydration, and some physiological and contextual aspects would help gain a better understanding of the relationship between personal psychological variables and the number of injuries sustained.

## Conclusions

In conclusion, this study shows that amateur long-distance runners with a high level of intrinsic motivation tend to suffer a greater number of injuries. However, no relationship was identified between external motivation and athletes' resilience and number of injuries. Psychological resilience was associated with a lower number of injuries by amateur runners.

## Supporting information

**S1 Dataset.**
(SAV)

## Acknowledgments

The authors wish to thank the Fortuna Sports Club as organizers of Behobia San Sebastian half marathon race, for allowing us to carry out this research.

## Author Contributions

**Conceptualization:** Patxi León-Guereño, Pedro Antonio Sánchez-Miguel.

**Data curation:** Patxi León-Guereño, Miguel Angel Tapia-Serrano.

**Formal analysis:** Patxi León-Guereño.

**Investigation:** Patxi León-Guereño, Miguel Angel Tapia-Serrano, Pedro Antonio Sánchez-Miguel.

**Methodology:** Patxi León-Guereño, Miguel Angel Tapia-Serrano, Pedro Antonio Sánchez-Miguel.

**Project administration:** Patxi León-Guereño.

**Resources:** Patxi León-Guereño.

**Software:** Miguel Angel Tapia-Serrano.

**Validation:** Patxi León-Guereño.

**Visualization:** Patxi León-Guereño, Miguel Angel Tapia-Serrano.

**Writing – original draft:** Pedro Antonio Sánchez-Miguel.

**Writing – review & editing:** Pedro Antonio Sánchez-Miguel.

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
