## [Decision Letter · Decision Letter 0]

4 Nov 2019

PONE-D-19-21930

The relationship of recreational runners’ motivation and resilience levels to the incidence of injury: A mediation model

PLOS ONE

Dear Dr. Tapia Serrano,

Thank you for submitting your manuscript to PLOS ONE. After careful consideration, we feel that it has merit but does not fully meet PLOS ONE’s publication criteria as it currently stands. Therefore, we invite you to submit a revised version of the manuscript that addresses the points raised during the review process.

We would appreciate receiving your revised manuscript by Dec 13 2019 11:59PM. To enhance the reproducibility of your results, we recommend that if applicable you deposit your laboratory protocols in protocols.io, where a protocol can be assigned its own identifier (DOI) such that it can be cited independently in the future. For instructions see: http://journals.plos.org/plosone/s/submission-guidelines#loc-laboratory-protocols

We look forward to receiving your revised manuscript.

Kind regards,

Daniel Boullosa

Academic Editor

PLOS ONE

Editorial requests

Please provide the anonymized datasets underlying Tables 1 and 2, this should include the individual data points used for the analyses.The cross-sectional design does not allow for causal relationships to be established, please ensure the language in the manuscript is revised to only refer to associations. Please also revise the conclusions section to ensure the claims are aligned to the data presented, for example, this fragment is unsupported as it implies a causal relationship while the design does not allow establishing a temporal sequence for the items measured ‘*The ability to overcome adverse situations reduces injury rate’.*In light of the cross-sectional design, it is also unclear that the data can support conclusions about mediating effects, which is one of the objectives outlined, please either revise to remove such a claim or provide a stronger justification for how this can be addressed based on the design and data collected.

Journal Requirements:

2. Under the Methods section please clearly indicate that participant consent was sought and type (written).

3. Please include your tables as part of your main manuscript and remove the individual files. Please note that supplementary tables (should remain/ be uploaded) as separate "supporting information" files

Reviewers' comments:

Reviewer's Responses to Questions

**Comments to the Author**

1. Is the manuscript technically sound, and do the data support the conclusions?

Reviewer #1: Partly

2. Has the statistical analysis been performed appropriately and rigorously? 

Reviewer #1: I Don't Know

3. Have the authors made all data underlying the findings in their manuscript fully available?

Reviewer #1: No

4. Is the manuscript presented in an intelligible fashion and written in standard English?

Reviewer #1: Yes

5. Review Comments to the Author

Reviewer #1: General

This paper concerns a cross-sectional study on the relationship between internal and external motivation, resilience and injury rates. It is an interesting and relevant study in a large group of runners that adds to the current literature. However, I have some concerns about the methods (injury definition, in- and exclusion criteria are missing) and privacy of the data (use of Google docs).

Additional comments

1. Is the manuscript technically sound, and do the data support the conclusions?

Partly, the data support the conclusions. However, in the conclusion a causal relationship is implied, while based on this cross-sectional study no causal relationships could be established. In addition, no injury definition is presented and it is therefore unclear how injury rate was calculated. Did you focus on acute or overuse injuries? Why ask for injuries during whole lifetime? This implies a long recall period.

2. Has the statistical analysis been performed appropriately and rigorously?

I am not sure, I am not familiar with process SPSS and mediation models. So, I would suggest to let an expert check these analyses and results.

3. Have the authors made all data underlying the findings in their manuscript fully available?

No, I couldn’t find links to the underlying datasets.

4. Is the manuscript presented in an intelligible fashion and written in standard English?

Yes

General comment: Throughout the manuscript ‘resilience’ is written in various ways, eg. ‘the capacity of sportspeople of overcoming adverse situations’ and. Please be consistent in your terminology.

Specific comments:

Abstract:

• The results of the first aim are not described.

• The statistical tests are not described, which makes it difficult to follow the results

• “(β = -0.062, CI= -0.137, 0.009)” Is the last number the p-value?

• In the discussion you describe the content of the discussion in the paper. However, I am curious about the actual content. What is described in this discussion, e.g. what are future research ideas / practical implications?

Introduction:

Line 67. The transition to the study of Van Gendt and studies on risk factors is a bit abrupt and needs more introduction.

Line 75. Which psychological attributes do you mean?

Line 80. This sentence is not complete. And why is motivational written with a capital (also line 82)?

Line 80-93. This paragraph contains a quite broad description of motivation in relation to running. Is that necessary? In my opinion it distracts the reader from your storyline. Potentially a change in the order of this paragraph or adding a few lines as introduction may help.

Line 124-131: The aims need some more introduction. Why study internal and external motivation in relation to injuries?

Line 127-131: In the abstract you describe another first aim. Which one is correct?

Line 132-139: I am a bit confused after reading the hypotheses, this may have to do with unclarity about the aims and unclarity about the dependent and independent variables? Please clarify.

Methods

Line 141-146. What were in- and exclusion criteria? A flow diagram may provide insight in the selection of runners for your analysis.

line 142. What do you mean with ‘ex post facto study’?

line 144-146. Years / years old. Please be consistent

Line 175-177. Information about how injury rate was calculated is missing. What was your injury definition? In addition, in the results no information about the duration or other characteristics of the injuries is presented.

Line 184-185. Google docs was used as survey tool: therefore I have some concerns about privacy of the participants, European regulations require stricter rules in regards to use of survey tools and the location where the data is saved, did you inform the runners about this?

Line 185-187. Please add some information about Fortuna club? How did you adjust the measurement tools? And why? To make them more appropriate for the target group?

Line 198-204. Why are you looking at differences between male and female runners in specific?

Line 219-221. What do you mean by ‘the remaining analyses’?

Results

Line 224-225. In table 1, the number of injuries is presented, however in the discussion you talk about injury rate. In my opinion, injury rate is a percentage and not a number. Adding an injury definition to the methods will provide clarity about your injury outcome.

Line 229-242. You describe ‘significant differences’, however I am not sure what differences you calculated. Also, I cannot find it in table 2.

Discussion

Line 280-283. This is a harsh statement. In my opinion, not presenting high levels of resilience does not automatically mean ‘less psychologically gifted’. Please rephrase.

Line 291, ‘to overcoming injury processes’ Do you mean recovery after injury/return to play?

Line 324-326 “In addition, controlling for training frequency and duration, as well as nutrition, hydration, and some physiological and contextual aspects would help to gain a better understanding of the relationship between personal psychological variables and the number of injuries sustained.”

How do you expect nutrition, hydration etc to influence the relation of motivation and injuries.

line 326-330. ”To conclude, an interesting line of research in the future would be to assess external motivation among amateur and professional athletes, in an attempt to see how external motivation differs according to the athletes’ level, their salary and the effect that television and social media have on them.” This comes a bit out of the blue for me. Please explain.

Conclusion

Line 336-339. Your conclusion that high resilience levels reduce injury rates implies a causal relationship. As you already described in the limitations section, causality cannot be established. Please rephrase.

Line 337. How could you improve resilience? Could you provide an example?

6. PLOS authors have the option to publish the peer review history of their article (what does this mean?). If published, this will include your full peer review and any attached files.

Reviewer #1: No

---

## [Author Response · Author response to Decision Letter 0]

10 Dec 2019

Cáceres, December, 10th 2019

Dear Daniel Boullosa,

The authors appreciate the time you devoted to reading our manuscript and helping us to craft an improved version of the investigation. We are pleased to clarify your concerns, which we believe have improved the quality and applicability of this work. Please, find below our responses to each of your observations. We have made a concerted attempt to systematically address the specific concerns raised for this revision and we have highlighted the alterations to this revision within the manuscript in red for your convenience.

Editor requests:

EDITOR: Please provide the anonymized datasets underlying Tables 1 and 2, this should include the individual data points used for the analyses.

AUTHORS: Thank you for your requirement. The authors are agreeing with contributions, so have decided to include the individual data points used for the analyses.

EDITOR: The cross-sectional design does not allow for causal relationships to be established, please ensure the language in the manuscript is revised to only refer to associations. Please also revise the conclusions section to ensure the claims are aligned to the data presented, for example, this fragment is unsupported as it implies a causal relationship while the design does not allow establishing a temporal sequence for the items measured ‘The ability to overcome adverse situations reduces injury rate’.

AUTHORS: We would like to thank reviewer for his comment in order to improve the quality of the manuscript. We have revised the language of the manuscript and we have removed causalities, using instead associations, in line with the cross-sectional design of the study. Lines:154-155, 313, 318-319, 365,366.

AUTHORS: We have revised the conclusion section as well, in order to ensure that the claims are aligned to the data presented: Line: 389-391: “Psychological resilience was associated with a lower number of injuries”.

EDITOR: In light of the cross-sectional design, it is also unclear that the data can support conclusions about mediating effects, which is one of the objectives outlined, please either revise to remove such a claim or provide a stronger justification for how this can be addressed based on the design and data collected.

AUTHORS: Thank you for your comment. The authors are agreeing with the contributions, so have decided to modify the description objectives by: evaluate the mediation role of intrinsic and extrinsic motivation on the number of injuries and resilience levels of amateur runners.

Reviewer(s)’ Comments to the AUTHORS: 

EDITOR: General

This paper concerns a cross-sectional study on the relationship between internal and external motivation, resilience and injury rates. It is an interesting and relevant study in a large group of runners that adds to the current literature. However, I have some concerns about the methods (injury definition, in- and exclusion criteria are missing) and privacy of the data (use of Google docs).

AUTHORS: Thank you for your comment. We added the definition of Injury (Lines:78-79) as follows “Injuries usually cause periods away from the sports practice”.

AUTHORS: Thank you for your comment. We added the exclusion criteria in design and participants of the study: Lines: 160-161 “using as exclusion criteria the athletes who did not finally take part at the race” [15] 

AUTHORS: Thank you for your comment. We added a more detailed information about privacy and informed consent used during data collection. Lines: 154-155. “all of them accepted the informed consent at the beginning of the survey”

Additional comments

Is the manuscript technically sound, and do the data support the conclusions?

EDITOR: Partly, the data support the conclusions. However, in the conclusion a causal relationship is implied, while based on this cross-sectional study no causal relationships could be established. In addition, no injury definition is presented and it is therefore unclear how injury rate was calculated. Did you focus on acute or overuse injuries? Why ask for injuries during whole lifetime? This implies a long recall period.

AUTHORS: Thanks a lot for your comments. We totally agree with reviewer. Thus, the causal relationship in the conclusion was removed and modified by associations, i.e.: “Psychological resilience was associated with a lower injury rate” (Lines: 389-391), and “which is associated at the same time with a lower number of injuries suffered by amateur runners” (lines: 392-393).

AUTHORS: Thank you for your comment. Injury definition was added in Lines 78-79. “Injuries usually cause periods away from the sports practice”, and injury rate was calculated asking athletes whether they were “injured” or “non injures”.

AUTHORS: Thanks a lot for your comments. Even though we did collect the data about the typology of injuries, we did not take this into account when analyzing the data in this research. It will be tested more specifically in future manuscripts, in fact we included in Lines: 377-377 “Another future research, following Quesada et. al [53] could be to analyse the factors associated with injury in recreational runners”

AUTHORS: Thank you for your comment. We did not ask about athletes’ injuries during the whole life, it would be definitely too long. We asked athletes about injuries during the last season, following different previous studies which support the idea to ask for injuries during the last twelve months Line 192: “gathering athletes’ number of injuries during the last twelve months” (Quesada, J. I. P., Kerr, Z. Y., Bertucci, W. M., & Carpes, F. P. (2019). A retrospective international study on factors associated with injury, discomfort and pain perception among cyclists. PloS one, 14(1), e0211197).

Have the authors made all data underlying the findings in their manuscript fully available?

EDITOR: No, I couldn’t find links to the underlying datasets.

AUTHORS: Thank you for your comment. We have included the data in the submission system in order to be consulted by reviewers and editor. 

Is the manuscript presented in an intelligible fashion and written in standard English?

Yes

EDITOR: General comment: Throughout the manuscript ‘resilience’ is written in various ways, eg. ‘the capacity of sportspeople of overcoming adverse situations’ and. Please be consistent in your terminology.

AUTHORS: Thank you for your comment. We used resilience or psychological resilience replacing different definitions of it in other to be consistent: Lines: 149-370. Zurita-Ortega, F., Chacón-Cuberos, R., Cofre-Bolados, C., Knox, E., and Muros, J. J. (2018). Relationship of resilience, anxiety and injuries in footballers: Structural equations analysis. PLoS ONE, 13, e0207860.https://doi.org/10.1371/journal.pone.0207860

EDITOR: Specific comments:

1. Abstract

EDITOR: The results of the first aim are not described.

AUTHORS: Thank you for your comment. We added the results of the first aim. Lines: 39-41. “Model 3 showed that significant differences were found with respect to resilience (p < 0.05) and intrinsic motivation (p < 0.05).” 

EDITOR: The statistical tests are not described, which makes it difficult to follow the results

AUTHORS: Thank you for your comment. We added the statistical test that we used in the Statistical analysis (page 9, line 225): "The association of number of injuries with resilience and motivations indicators was tested by linear regression model. Linear regression analyses were hierarchically adjusted for covariates by creating 3 models. Model 1 further adjusted for by age, sex and resilience; Model 2 was additionally adjusted for intrinsic motivation, and Model 3 further adjusted for extrinsic motivation. 

The PROCESS SPSS version 3.00 Macro (Hayes, 2017) was used for the mediation analysis between number of injuries and resilience. Two different models were established. For mediation model one, extrinsic motivation was specified as a mediating variable; resilience was used as an independent variable and the number of injuries was used as a dependent variable. "

EDITOR: “(β = -0.062, CI= -0.137, 0.009)” Is the last number the p-value?

AUTHORS: Thank you for your comment. Regarding the question indicated, that value is the one corresponding to the confidence interval, resulting from the mediation analysis. If this interval does not pass the value of 0, it is therefore considered that the indirect effect is significant.

EDITOR: In the discussion you describe the content of the discussion in the paper. However, I am curious about the actual content. What is described in this discussion, e.g. what are future research ideas / practical implications?

AUTHORS: Thank you for your comment. For instance, we find that intrinsic motivation is associated with a higher incidence of injury. Therefore, it would be important that very intrinsically motivated amateur athletes, having a good training plan, and being leaded by professionals, so that, the association between their intrinsic motivation and incidence of injury will be weaker, even when they push themselves hard.

2. Introduction

EDITOR: Line 67. The transition to the study of Van Gendt and studies on risk factors is a bit abrupt and needs more introduction.

AUTHORS: Thank you for your comment. We added a little introduction to Van Gendt. Line 72: “In this direction”

EDITOR: Line 75. Which psychological attributes do you mean?

AUTHORS: Thank you for your comment. We added couple of examples of psychological attributed that have been analyzed related to athletes’ injuries. Line 82: “e.g. stress, resilience”

EDITOR: Line 80. This sentence is not complete. And why is motivational written with a capital (also line 82)?

AUTHORS: Thank you for your comment. We completed the sentence Line: 87,88 “characteristics of athletes” and we change the capital letter of motivation in lines 87 and 91.

EDITOR: Line 80-93. This paragraph contains a quite broad description of motivation in relation to running. Is that necessary? In my opinion it distracts the reader from your storyline. Potentially a change in the order of this paragraph or adding a few lines as introduction may help.

AUTHORS: Thank you for your comment. We added a little introduction in other to improve reader’s comprehension. Lines 88-89. “with several studies based on Self-Determination Theory (SDT) (Deci and Ryan, 2000)” 

EDITOR: Line 124-131: The aims need some more introduction. Why study internal and external motivation in relation to injuries?

AUTHORS: Thank you for your comment. We added some more introduction in other to improve reader understanding. Lines: 136-139. “However, sportsmen injuries have not been associated yet with athletes’ motivational characteristics, or athletes’ internal and external motivations.”

EDITOR: Line 127-131: In the abstract you describe another first aim. Which one is correct?

AUTHORS: Thank you for your comment. We adapted the first objective considering the objective stablished in the abstract. Lines: 141-143. “to examine the relationship between runners’ resilience levels, internal and external motivation and incidence of injury”

EDITOR: Line 132-139: I am a bit confused after reading the hypotheses, this may have to do with unclarity about the aims and unclarity about the dependent and independent variables? Please clarify.

AUTHORS: Thank you for your comment. We try to give you a clearer explanation in the following lines: The first hypothesis is that the number of injuries will be associated with the psychological resilience of athletes, i.e., the higher athlete’s resilience, the better athletes will deal with stressful situations, so, the association with the incidence of injury it will be lower. In the second hypothesis, our hypothesis connects athletes’ intrinsic and extrinsic motivation with the association with athletes’ injury rate, guessing that intrinsic motivation will be more associate with this incidence than extrinsic motivation. Lines. 147-155.

3. Methods

EDITOR: Line 141-146. What were in- and exclusion criteria? A flow diagram may provide insight in the selection of runners for your analysis.

AUTHORS: Thank you for your comment. Los autores están de acuerdo con las aportaciones de los revisores y han decidido incluir un diagrama de flujo con los criterios de selección de los participantes para la presente investigación.

EDITOR: line 142. What do you mean with ‘ex post facto study’? 

AUTHORS: Thank you for your comment. Ex post facto study mean “retroactive” or after the fact, and as in our case the data collection was carried out two weeks before the race, we changed, and we decided adding “Cross sectional design”, which we consider more suitable for this research (Line: 158,159) 

EDITOR: line 144-146. Years / years old. Please be consistent

AUTHORS: Thank you for your comment. In order to be consistent, we decided using “years” instead of “years old” (Lines: 163,164).

EDITOR: Line 175-177. Information about how injury rate was calculated is missing. What was your injury definition? In addition, in the results no information about the duration or other characteristics of the injuries is presented.

AUTHORS: Thank you for your comment. We added how injury rate was calculated “Injured “or “Non Injured” were labelled at the time of the data collection” Lines: 191,192.

AUTHORS: Thank you for your comment. We define what injury mean for us in Lines 80,81. “Injuries usually cause periods away from the sports practice”

AUTHORS: Thank you for your comment. Even though we collected the data about typology of injuries, we decided not including it in this research, because It was not connected with the main objectives of dis research.

EDITOR: Line 184-185. Google docs was used as survey tool: therefore, I have some concerns about privacy of the participants, European regulations require stricter rules in regards to use of survey tools and the location where the data is saved, did you inform the runners about this?

AUTHORS: Thank you for your comment. We added an extra information specifying that informed consent was asked at the beginning of the questionnaire. Lines: 164,165. “all of them accepted the informed consent at the beginning of the survey”

EDITOR: Line 185-187. Please add some information about Fortuna club? How did you adjust the measurement tools? And why? To make them more appropriate for the target group?

AUTHORS: Thank you for your comment. We added that Fortuna is an “Sports club” Line 206. Apart from that, we adapted the survey considering questions of interest for Fortuna Sports Club, since they are involved in a project to boost women participation in future editions. This way, we tried to adjust and make the tool more appropriate for the target group.

EDITOR: Line 198-204. Why are you looking at differences between male and female runners in specific?

AUTHORS: Thank you for your comment. We were looking at differences between male and female athletes since women’s participation is significantly lower at this race, and the number of research is lower for women [3].

[3] Mujika Alberdi, A., García Arrizabalaga, I., and Gibaja Martíns, J. J. (2018). Impact of the behobia-san sebastián race on promoting healthy lifestyles. [Incidencia de la carrera Behobia-San Sebastián en el fomento de estilo de vida saludable] Apunts.Educacion Fisica Y Deportes, (131), 34-48. doi:10.5672/apunts.2014-0983.es.(2018/1).131.03

EDITOR: Line 219-221. What do you mean by ‘the remaining analyses’?

AUTHORS: Thank you for your comment. With remaining statistical analyses (Line: 250), we mean the rest of the analyses that we carried out during this research: descriptive statistics, student’s t test, and the chi-square... Lines: 220,221

4. Results

EDITOR: Line 224-225. In table 1, the number of injuries is presented, however in the discussion you talk about injury rate. In my opinion, injury rate is a percentage and not a number. Adding an injury definition to the methods will provide clarity about your injury outcome.

AUTHORS: Thank you for your comment. We changed “injury rate” and we used instead “number of injuries” in the whole manuscript, in order to be coherent with the table 1 and to be consistent at the same time., Lines: 37, 311, 318, 314, 342, 372-387-388.

EDITOR: Line 229-242. You describe ‘significant differences’, however I am not sure what differences you calculated. Also, I cannot find it in table 2.

AUTHORS: Thank you for your comment. The authors agree with your contributions, so we have decided to modify the description (Please, see page 11, line 260-265): "Table 2 shows the associations between number of injuries and resilience and motivational indicators. In model 1, the number of injuries was not positively associated with resilience levels after controlling for sex, age (β = -0.063; p > 0.05). In Model 2 the positive association between number of injuries and intrinsic motivation remained significant (β = 0.123; p < 0.05). Finally, in Model 3 the number of injuries was relationship significant and positively with extrinsic motivation (β = 0.140; p < 0.05)."

5. Discussion

EDITOR: Line 280-283. This is a harsh statement. In my opinion, not presenting high levels of resilience does not automatically mean ‘less psychologically gifted’. Please rephrase.

AUTHORS: Thank you for your comment. We rephrased the sentence, and we wrote instead: “athletes with lower level of psychological resilience” Lines: 328,329.

EDITOR: Line 291, ‘to overcoming injury processes’ Do you mean recovery after injury/return to play?

AUTHORS: Thank you for your comment. We extended the information in order to improve text’s meaning. We added “, i.e. recovery after injury” Line 338.

EDITOR: Line 324-326 “In addition, controlling for training frequency and duration, as well as nutrition, hydration, and some physiological and contextual aspects would help to gain a better understanding of the relationship between personal psychological variables and the number of injuries sustained.”

How do you expect nutrition, hydration etc to influence the relation of motivation and injuries.

AUTHORS: Thank you for your comment. We do not expect that nutrition, hydration or another variable to influence directly in the relation between motivation and injuries, but those variables could be directly related to the incidence of injury of athletes, and controlling them, we could get a more accurate information about the association between the number of injuries and motivational aspects of amateur runners. Line: 373-376.

EDITOR: line 326-330. “To conclude, an interesting line of research in the future would be to assess external motivation among amateur and professional athletes, in an attempt to see how external motivation differs according to the athletes’ level, their salary and the effect that television and social media have on them.” This comes a bit out of the blue for me. Please explain.

AUTHORS: Thank you for your comment. Seeing that contextual aspects do have influence on athletes’ personal variables. Line: 116 [31,32] (Hosseini and Besharat, 2010; Sarkar and Fletcher, 2014). Line: 125,126 [41,42] (Pedro and Veloso, 2018; Wagstaff, Chris, Hings, Larner, and Fletcher, 2018). Looking ta future research lines, we would like to link and asses till what extend, some of these contextual variables can effect on the association between athlete’s motivation and their number of injuries, since the number of injuries of athletes can be associated to athletes’ stress levels, and to their psychological resilience. Line: 376

6. Conclusion

EDITOR: Line 336-339. Your conclusion that high resilience levels reduce injury rates implies a causal relationship. As you already described in the limitations section, causality cannot be established. Please rephrase.

AUTHORS: Thank you for your comment. We rephrased the sentences, using association instead of causality, thus being coherent with this cross sectional study. Lines: 389, 390

EDITOR: Line 337. How could you improve resilience? Could you provide an example?

AUTHORS: Thank you for your comment. We changed resilience improvement by “being more resilient”, since psychological interventions can be planned with high performance athletes in order to improve this personal variable (Codonhato, R., Rubio, V., Oliveira, P. M. P., Resende, C. F., Rosa, B. A. M., Pujals, C., and Fiorese, L. 2018). In our case, as we are analysing amateur athletes, we considered mor appropriate to talk about just resilience level of participants. Line: 388. 

Finally, and after having made all the modifications and comments discussed above, the authors appreciate the contributions made by the reviewers, as they help us improve the quality of the article. We hope that the comments regarding the contributions of the reviewers have been made correctly and that the work may be of interest for later publication. 

Please, if further changes are needed, just let us know and we would try to correct them as soon as possible.

We thank you in advance for your attention and treatment,

Sincerely

Dr. Pedro Antonio Sánchez Miguel

---

## [Decision Letter · Decision Letter 1]

10 Feb 2020

PONE-D-19-21930R1

The relationship of recreational runners’ motivation and resilience levels to the incidence of injury: A mediation model

PLOS ONE

Dear Dr. Tapia Serrano,

Thank you for submitting your manuscript to PLOS ONE. After careful consideration, we feel that it has merit but does not fully meet PLOS ONE’s publication criteria as it currently stands. Therefore, we invite you to submit a revised version of the manuscript that addresses the points raised during the review process.

Please, address all the concerns of the reviewers and, if it is not possible to sufficiently address all their concerns, assume them as limitations of the study in a specific paragraph in the discussion.

We would appreciate receiving your revised manuscript by Mar 26 2020 11:59PM. To enhance the reproducibility of your results, we recommend that if applicable you deposit your laboratory protocols in protocols.io, where a protocol can be assigned its own identifier (DOI) such that it can be cited independently in the future. For instructions see: http://journals.plos.org/plosone/s/submission-guidelines#loc-laboratory-protocols

We look forward to receiving your revised manuscript.

Kind regards,

Daniel Boullosa

Academic Editor

PLOS ONE

Reviewers' comments:

Reviewer's Responses to Questions

**Comments to the Author**

1. If the authors have adequately addressed your comments raised in a previous round of review and you feel that this manuscript is now acceptable for publication, you may indicate that here to bypass the “Comments to the Author” section, enter your conflict of interest statement in the “Confidential to Editor” section, and submit your "Accept" recommendation.

Reviewer #1: (No Response)

Reviewer #2: (No Response)

2. Is the manuscript technically sound, and do the data support the conclusions?

Reviewer #1: Yes

Reviewer #2: No

3. Has the statistical analysis been performed appropriately and rigorously? 

Reviewer #1: I Don't Know

Reviewer #2: Yes

4. Have the authors made all data underlying the findings in their manuscript fully available?

Reviewer #1: Yes

Reviewer #2: Yes

5. Is the manuscript presented in an intelligible fashion and written in standard English?

Reviewer #1: Yes

Reviewer #2: Yes

6. Review Comments to the Author

Reviewer #1: Thank you for addressing all the comments. In my opinion the changes you made in this manuscript lead to a good qualitative study. I have a few small comments left.

AUTHORS: Thank you for your comment. We were looking at differences between male

and female athletes since women’s participation is significantly lower at this race, and the

number of research is lower for women [3].

[3] Mujika Alberdi, A., García Arrizabalaga, I., and Gibaja Martíns, J. J. (2018). Impact

of the behobia-san sebastián race on promoting healthy lifestyles. [Incidencia de la

carrera Behobia-San Sebastián en el fomento de estilo de vida saludable]

Apunts.Educacion Fisica Y Deportes, (131), 34-48. doi:10.5672/apunts.2014-

0983.es.(2018/1).131.03

Please also mention this paper and explanation in the manuscript

AUTHORS: Thank you for your comment. For instance, we find that intrinsic motivation

is associated with a higher incidence of injury. Therefore, it would be important that very

intrinsically motivated amateur athletes, having a good training plan, and being leaded by

professionals, so that, the association between their intrinsic motivation and incidence of

injury will be weaker, even when they push themselves hard

Thank you for the explanation. Could you please change this in the abstract as well?

Reviewer #2: The cross-sectional manuscript The relationship of recreational runners’ motivation and resilience levels to the incidence of injury: A mediation model concerns the relationship between resilience, motivation and injury rates. It is a current topic of interest in sports science and their writing, sample size and study-desing are generally sound. However, I recommend it to be rejected for publication for the following reasons:

The correlations observed in the study are not scientifically explained. Scientific evidence is lacking to support the assumption that matters in the present article (the enlightenment of how these physiological variables relate to injuries). Correlation does not mean causality. In my humble opinion, the discussion section does not meet the requirements and high standards of quality for a prestigious journal such as PlosOne.

7. PLOS authors have the option to publish the peer review history of their article (what does this mean?). If published, this will include your full peer review and any attached files.

Reviewer #1: No

Reviewer #2: No

---

## [Author Response · Author response to Decision Letter 1]

17 Feb 2020

Cáceres, February, 17th 2020

Dear Daniel Boullosa,

The authors appreciate the time you spent to reading our manuscript and helping us to craft an improved version of the investigation. We are pleased to clarify your concerns, which we believe have improved the quality and applicability of this work. Please, find below our responses to each of your observations. We have made a concerted attempt to systematically address the specific concerns raised for this revision and we have highlighted the alterations to this revision within the manuscript in red for your convenience.

REVIEWER: “Thank you for your comment. We were looking at differences between male and female athletes since women’s participation is significantly lower at this race, and the number of research is lower for women [3]. Mujika Alberdi, A., García Arrizabalaga, I., and Gibaja Martíns, J. J. (2018). Impact of the behobia-san sebastián race on promoting healthy lifestyles. [Incidencia de la carrera Behobia-San Sebastián en el fomento de estilo de vida saludable] Apunts.Educacion Fisica Y Deportes, (131), 34-48. doi:10.5672/apunts.2014-0983.es.(2018/1).131.03.” Please also mention this paper and explanation in the manuscript.

AUTHORS: Thank you so much for your comment. We mentioned this paper (Lines: 47, 103) and we added the explanation in Lines: 102-104 “,and taking into account that women’s participation is significantly lower at the Behobia-San Sebastian race, and the number of research is lower for women [3] differences between male and female athletes will be considered”.

REVIEWER: “For instance, we find that intrinsic motivation

is associated with a higher incidence of injury. Therefore, it would be important that very

intrinsically motivated amateur athletes, having a good training plan, and being leaded by

professionals, so that, the association between their intrinsic motivation and incidence of

injury will be weaker, even when they push themselves hard”

Thank you for the explanation. Could you please change this in the abstract as well?

AUTHORS: We appreciate very mucho this indication and therefore, we have changed the information in the abstract section, and we have added the following information: “, i.e., it was found that intrinsic motivation was associated with a higher incidence of injury,” Lines 37,38, and in Lines 42-45: “Moreover, it is considered important that very intrinsically motivated amateur athletes, having a good training plan, and being leaded by professionals, therefore, the association between their intrinsic motivation and incidence of injury would be weaker, even when they push themselves hard”, completing thus the abstract.

REVIEWER: The cross-sectional manuscript. The relationship of recreational runners’ motivation and resilience levels to the incidence of injury: A mediation model concerns the relationship between resilience, motivation and injury rates. It is a current topic of interest in sports science and their writing, sample size and study-desing are generally sound. However, I recommend it to be rejected for publication for the following reasons:

The correlations observed in the study are not scientifically explained. Scientific evidence is lacking to support the assumption that matters in the present article (the enlightenment of how these physiological variables relate to injuries). Correlation does not mean causality. In my humble opinion, the discussion section does not meet the requirements and high standards of quality for a prestigious journal such as PlosOne.

AUTHORS: Thank you for your comment. Since the research line is very new, we are afraid that there is not easy to support scientific evidence about the obtained results in the literature, however, we added a paragraph in the limitation section. Lines: 291-293 “Likewise, another limitation was that scientific evidence was lacking due to the novelty of this topic, and physiological variables related to injuries were not considered." Previous research had associated psychological variables such as resilience or anxiety, or competitive level with athletes’ injuries [16,41], and these investigations have been considered during this manuscript.

16.Zurita-Ortega F, Chacón-Cuberos R, Cofre-Bolados C, Knox E, Muros JJ. Relationship of resilience, anxiety and injuries in footballers: Structural equations analysis. Ewen HH, editor. PLoS One. 2018;13: e0207860. doi:10.1371/journal.pone.0207860

44.Castro Sánchez M, Chacón Cuberos R, Zurita-Ortega F, Espejo Garcés T. Levels of resilience based on sport discipline, competitive level and sport injuries. RETOS-Nuevas Tendencias En Educ Fis Deport Y Recreacion. 2016; 162–165. 

On other hand, we agree that correlation does not mean causality, in fact, we have corrected the manuscript in the previous review, changing causalities for associations, “We have revised the language of the manuscript and we have removed causalities, using instead associations, in line with the cross-sectional design of the study.” e.g. Lines:154-155, 313, 318-319, 365,366. Nevertheless, and following your concern, the cross-sectional design was considered a limitation of the study. Lines 291-293 “The cross-sectional study design was another limitation, as it did not allow causal relationships between the study variables to be established”.

Regarding the discussion section, we do not understand what we are supposed to improve, since we used the latest publications in prestigious journals like PLOS One, and we followed the way the discussion is organized in them. E.g: 12. van der Worp MP, ten Haaf DSM, van Cingel R, de Wijer A, Nijhuis-van der Sanden MWG, Staal JB. Injuries in Runners; A Systematic Review on Risk Factors and Sex Differences. Zadpoor AA, editor. PLoS One. 2015;10: e0114937. doi:10.1371/journal.pone.0114937, 16.Zurita-Ortega F, Chacón-Cuberos R, Cofre-Bolados C, Knox E, Muros JJ. Relationship of resilience, anxiety and injuries in footballers: Structural equations analysis. Ewen HH, editor. PLoS One. 2018;13: e0207860. doi:10.1371/journal.pone.0207860, or 43. Codonhato R, Rubio V, Oliveira PMP, Resende CF, Rosa BAM, Pujals C, et al. Resilience, stress and injuries in the context of the Brazilian elite rhythmic gymnastics. Bergamini E, editor. PLoS One. 2018;13: e0210174. doi:10.1371/journal.pone.0210174

With respect to correlations comments’, the author would like to thank reviewers´ contributions, and we have decided to modify the redaction of correlations by: “In model 1, the number of injuries was not significant associated with resilience levels after being controlled for sex and age (β = -0.063; p = 0.089). However, in model 2 the number of injuries was positively associated with intrinsic motivation after being controlled for sex, age and resilience (β = 0.123; p < 0.05). In the other hand, there were not significant associations between the number of injuries and extrinsic motivation after the additional adjustment for intrinsic motivation in model 3 (β = -0.102; p = 0.072)”.

Thus, we hope that the information added is consistent with a scientific explanation. Please, if further information is needed, let us know, and we will attempt to include more scientific information.

Finally, and after having made all the modifications and comments discussed above, the authors appreciate the contributions developed by the reviewers, as they have helped us to improve the quality of the article. We hope that the comments regarding the contributions of the reviewers have been conducted correctly and the work may be of interest for later publication. 

Please, if further changes are needed, just let us know and we would try to correct them as soon as possible.

We thank you in advance for your consideration,

Sincerely

Dr. Pedro Antonio Sánchez Miguel

---

## [Editor Report · Decision Letter 2]

30 Mar 2020

The relationship of recreational runners’ motivation and resilience levels to the incidence of injury: A mediation model

PONE-D-19-21930R2

Dear Dr. Tapia Serrano,

We are pleased to inform you that your manuscript has been judged scientifically suitable for publication and will be formally accepted for publication once it complies with all outstanding requirements.

Within one week, you will receive an e-mail containing information on the amendments required prior to publication. While addressing those items, please also address the editorial requests outlined below.

When all required modifications have been addressed, you will receive a formal acceptance letter and your manuscript will proceed to our production department and be scheduled for publication.

With kind regards,

Daniel Boullosa

Academic Editor

PLOS ONE

Editorial requests

Abstract – please remove references to 'effect' from the abstract, given that the study can only establish associations. Also the last fragment in the abstract is unclear and the potential benefits of having a trainer have not been evaluated in the study, please revise that fragment for clarity and to ensure conclusions align to what can be supported by the data.

Please carefully check the written language one more time to ensure it is at publication standard.

Lines 226-227 Please revise ‘*In model 1, the number of injuries did not associate significant with resilience levels after controlling for sex and age*’ to ‘*In model 1, the number of injuries did not show a significant association with resilience levels after controlling for sex and age’*

Lines 229-231 – this fragment is unclear please revise it for clarity ‘I*n the other hand, the did not find associations significant between number of injuries and extrinsic motivation after additional adjustment for intrinsic motivation in model 3’*

Line 305 ‘ this fragment is unclear, recommend deleting the fragment ‘*scientific evidence was lacking due to the novelty of this topic’*

Discussion – please include among the limitations that the information on injuries was self-reported.

Lines 320-326 this paragraph appears to be duplicated, also the last sentence in the paragraph follows a circular argument, can that be revised or deleted.

---

## [Editor Report · Acceptance letter]

20 Apr 2020

PONE-D-19-21930R2 

The relationship of recreational runners’ motivation and resilience levels to the incidence of injury: A mediation model 

Dear Dr. Tapia Serrano:

I am pleased to inform you that your manuscript has been deemed suitable for publication in PLOS ONE. Congratulations! Your manuscript is now with our production department. 

With kind regards,

on behalf of

Dr. Daniel Boullosa 

Academic Editor

PLOS ONE